# The effect of crystallite size on pressure amplification in switchable porous solids

Simon Krause [1], Volodymyr Bon [1], Irena Senkovska [1], Daniel M. Többens[2], Dirk Wallacher[3], Renjith S. Pillai [4], Guillaume Maurin [4] & Stefan Kaskel [1]

Negative gas adsorption (NGA) in ordered mesoporous solids is associated with giant contractive structural transitions traversing through metastable states. Here, by systematically downsizing the crystal dimensions of a mesoporous MOF (DUT-49) from several micrometers to less than 200 nm, counterintuitive NGA phenomena are demonstrated to critically depend on the primary crystallite size. Adsorbing probe molecules, such as *n*-butane or nitrogen, gives insights into size-dependent activation barriers and thermodynamics associated with guest-induced network contraction. Below a critical crystal size, the nitrogen adsorption-induced breathing is completely suppressed as detected using parallelized synchrotron X-ray diffraction–adsorption instrumentation. In contrast, even the smallest particles show NGA in the presence of *n*-butane, however, associated with a significantly reduced pressure amplification. Consequently, the magnitude of NGA in terms of amount of gas expulsed and pressure amplification can be tuned, potentially paving the way towards innovative concepts for pressure amplification in micro- and macro-system engineering.

[1] Department of Inorganic Chemistry, Technische Universität Dresden, Bergstrasse 66, 01062 Dresden, Germany. [2] Department Structure and Dynamics of Energy Materials, Helmholtz-Zentrum Berlin für Materialien und Energie, Hahn-Meitner-Platz 1, 14109 Berlin, Germany. [3] Department Sample Environments, Helmholtz-Zentrum Berlin für Materialien und Energie, Hahn-Meitner-Platz 1, 14109 Berlin, Germany. [4] Institut Charles Gerhardt Montpellier UMR 5253 CNRS UM ENSCM, Université Montpellier, Place E. Bataillon, 34095, Montpellier Cedex 05, France. Correspondence and requests for materials should be addressed to S.K. (email: stefan.kaskel@chemie.tu-dresden.de)

The discovery of metal–organic frameworks (MOFs) in the late 1990s represents a major breakthrough in terms of rational design and molecularly defined tailoring of adsorbents to achieve specific functions[1]. By today, several thousand structures[2] have been synthesized in the laboratory, produced by industry, or even found in nature[3], among them record holders in terms of specific surface area, storage capacity[4] and separation selectivity[5] for energy carriers such as methane and also greenhouse gases such as $CO_2$[6] and toxic gases[7]. A limited number of MOFs show previously unprecedented adsorption phenomena such as gating, flexing and breathing upon guest adsorption and desorption caused by dynamic structural transformations[8–11]. Porosity switching, i.e., the ability to dynamically adjust pore sizes according to the guest shape and size, culminating in guest-specific pore opening, is a unique feature among porous materials with promising applications in the areas of storage, selective sensing and separation technology[12–14]. In particular in the field of gas storage, gating MOFs are recently recognized to deliver the almost ideal working capacity because they close their pores after reaching a lower critical pressure resulting in a perfect gas tank discharge[13]. Because of the crystalline character, MOFs are ideal candidates to achieve a deeper understanding of adsorption-induced structural transformations and host–guest interactions[15] on both theoretical[14,16] and experimental[17,18] levels in particular by developing advanced diffraction techniques[19–22].

Recently, we have discovered negative gas adsorption (NGA), a new adsorption phenomenon, upon methane adsorption in DUT-49 (Dresden University of Technology (DUT)) which is the first example of a pressure-amplifying material[23]. The network is assembled from porous cuboctahedral metal–organic polyhedra (MOPs, $d = 1.0$ nm) connected via biphenylene struts into a face-centered cubic net with 1.7 and 2.4 nm wide interpolyhedral voids.

Upon isothermal adsorption of methane at 111 K, DUT-49 shows a massive contraction and subsequent spontaneous release of methane as a reaction towards a defined stimulus, resulting in an overall pressure increase in a closed volume[23]. This counterintuitive pressure amplification (PA) is associated with an unusual, negative step in the adsorption isotherm, a characteristic signature of NGA, followed by a wide hysteresis associated with breathing. Combining molecular simulations with in situ powder X-ray diffraction (PXRD) and spectroscopic experiments revealed a metastable overloaded state of DUT-49op (open pore (op)) folding via a colossal structural contraction into DUT-49cp (contracted pore (cp)) associated with molecule ejection to be the origin of NGA[23,24].

The driving force for the contraction is energetic with a negligible entropic contribution[24]. The contraction of empty DUT-49op is an endothermic process. However, the magnitude of the methane adsorption enthalpy $|\Delta H|$ gain is higher for the contracted phase (DUT-49cp) due to the smaller pores ($d_{cp} < 1.0$ nm) compared with DUT-49op, and hence induces the contraction by compensating the energy required for the empty host compression as demonstrated in our earlier work[23]. NGA is not a singular phenomenon for methane alone but has been recently observed in DUT-49 for other gases such as $n$-butane and xenon in a temperature range close to their respective boiling points[25]. However, astonishingly upon adsorption of nitrogen at 77 K, a type Ib isotherm, as it is typical for rigid architectures, was initially reported without structural contraction[26]. The latter observation indicates a more subtle balance of adsorption energetics for nitrogen which allows for probing changes of the framework energetics in detail. This motivated us to accurately probe cooperative network energetics against adsorption-induced breathing and NGA behavior by tailoring the crystal size via

adjusted particle size distributions. For a rational development of next generations of NGA materials enabling high-pressure amplification, an understanding of critical size phenomena affecting metastability and NGA is essential.

Here we describe the entire suppression of NGA and breathing in a pressure-amplifying material below a critical particle size. Consequently, we propose crystal size tuning of an NGA material as a technique to modulate PA.

## Results

**Size controlled preparation of DUT-49 crystals**. To analyze the effects of crystallite size, a series of 16 DUT-49 samples was synthesized by adjusting the reaction conditions. The achieved tailored crystal size distributions cover a range of average crystal sizes from 100 nm to 11 μm (Supplementary Tables 1–3, samples are denoted as DUT-49(**1–16**)). While the ligand to metal salt ratio was kept constant during the synthesis, varying linker concentrations as well as addition of acetic acid and trimethylamine were used to control the crystal dimensions. Dilution of the reaction mixture as well as the addition of varying amounts of acetic acid as modulator led to an increase of the crystal size, a rational approach frequently used in the synthesis of Zr-MOF single crystals[27]. By using saturated linker solutions as well as adding triethylamine as a base, the crystals were downsized to less than 200 nm. All samples were identically desolvated using an advanced protocol involving supercritical $CO_2$ extraction followed by thermal activation in dynamic vacuum[17,26]. Crystal size distributions were determined by scanning electron microscopy (SEM) analysis (the measured size corresponds to the edge length of the cubic crystals, Fig. 1).

Further details about synthetic conditions are provided in the Methods section. Sample analyses such as SEM, PXRD, nitrogen adsorption at 77 K and thermogravimetric analysis (TGA) (Supplementary Fig. 1–20), selected diffuse reflectance infrared Fourier transform (DRIFT) spectra (Supplementary Fig. 23) and elemental analysis (EA) (Supplementary Table 5) are provided in the Supplementary Information, respectively. The analyses of the synthesized and activated samples by PXRD, TGA, EA, nuclear magnetic resonance spectroscopy (NMR) and DRIFT show no significant variations in composition, indicating that all samples correspond to the desired pure DUT-49 phase.

**Crystallite size dependence of adsorption-induced network contraction and NGA**. Nitrogen physisorption measurements at 77 K reveal a clear impact of particle size on structural switching behavior in DUT-49 (Fig. 1c, f). Samples of DUT-49(**1–8**) synthesized with increasing amounts of acetic acid show a pronounced hysteresis and NGA indicating a structural contraction induced by nitrogen at 77 K (Supplementary Fig. 1–8), which is fully supported by advanced parallelized diffraction/adsorption investigations (Fig. 2).

In contrast, samples DUT-49(**14–16**) with crystal size below 1 μm show no hysteresis (Fig. 1f). All other samples with an average crystal size above 1 μm show widening of the hysteresis with increasing average crystal size (Supplementary Fig. 9–13). In addition, NGA steps are observed for samples that show a hysteresis, suggesting that a structural transition comparable to those previously observed for other gases (such as methane and $n$-butane) takes place; however, for nitrogen it is highly dependent on the crystal size. Moreover, the total nitrogen uptake increases with increasing crystal size from 48 mmol g$^{-1}$ for DUT-49(**16**) (107 nm) to 79 mmol g$^{-1}$ for DUT-49(**2**) (8.89 μm). In parallel, the NGA step decreases from 9.66 mmol g$^{-1}$ for DUT-49(**3**) (8.75 μm) to 0.22 mmol g$^{-1}$ for DUT-49(**10**) (1.26 μm) and vanishes in samples below average crystal sizes of 1 μm

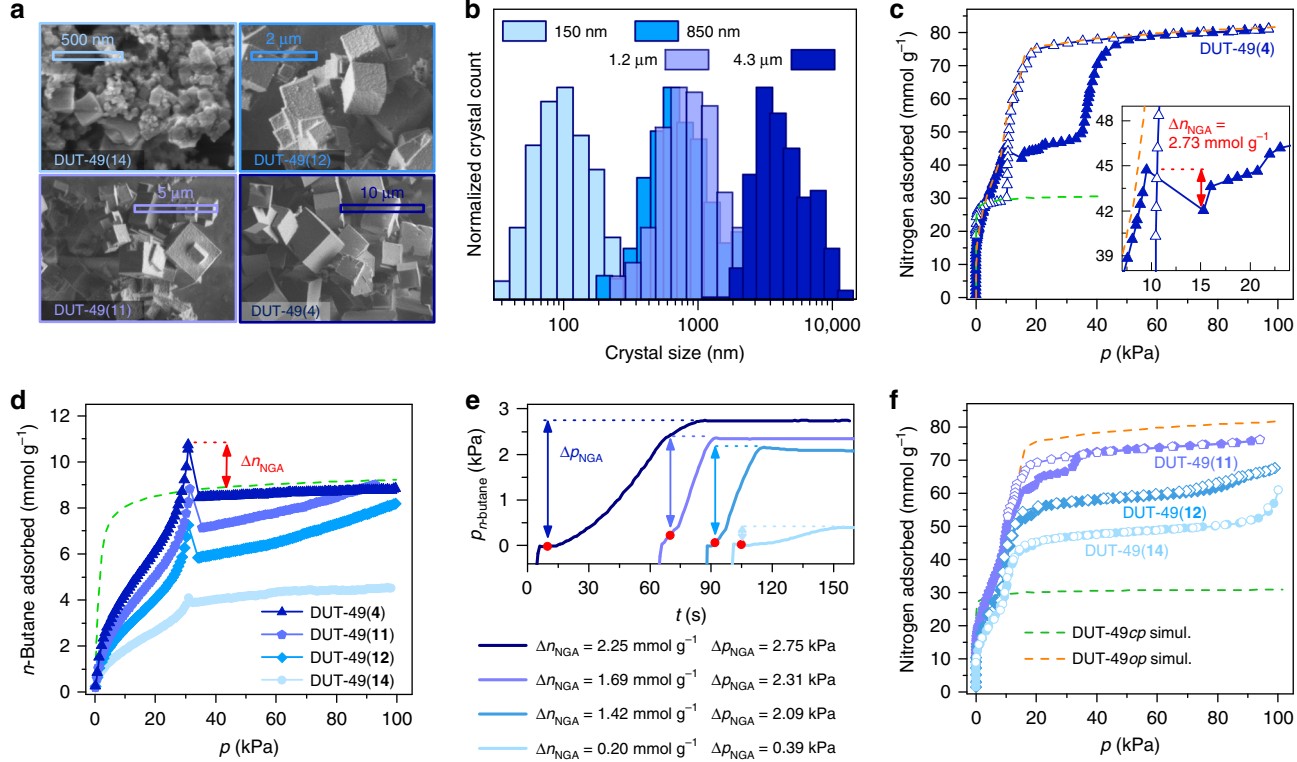

**Fig. 1** Nitrogen and *n*-butane adsorption isotherms of selected samples with different crystal size distribution. **a** SEM images of DUT-49 crystals with different size distributions, scale bars (500 nm top left, 2 μm top right, 5 μm bottom left, 10 μm bottom right) and sample codes are indicated in the Figure. **b** Normalized crystal size distribution of samples represented in SEM images. **c** Nitrogen physisorption isotherm at 77 K of DUT-49(**4**) (average crystal size 4.3 μm, inset shows enlargement of NGA region). **d** *n*-Butane adsorption isotherms at 298 K. **e** Corresponding kinetic profile upon NGA with values for $\Delta n_{NGA}$, pressure evolution in the cell upon NGA during *n*-butane adsorption at 298 K. **f** Nitrogen physisorption isotherms at 77 K of DUT-49(**11**), DUT-49 (**12**) and DUT-49(**14**) with average crystal size of 1.2 μm, 850 nm and 150 nm, respectively. color code: DUT-49(**4**) dark blue triangles, DUT-49(**11**) purple pentagons, DUT-49(**12**) light blue diamonds, DUT-49(**14**) turquoise circles. Closed symbols correspond to adsorption and empty symbols to desorption. Orange and green dashed lines correspond to the simulated isotherms of DUT-49*op* and DUT-49*cp*, respectively. Red marks indicate NGA transition

(Supplementary Table 4). The representative experiments clearly indicate a correlation of nitrogen adsorption-induced NGA at 77 K and the crystal size. The critical crystal size for nitrogen-induced NGA transitions at 77 K is estimated to be in the range of 1–2 μm.

To understand if this critical size is a characteristic feature of nitrogen gas, we have analyzed the response towards *n*-butane as an alternative probe molecule at 298 K, a temperature and pressure range that is relevant for the integration of switchable MOFs into micropneumatic devices. From the 16 synthesized samples, the 4 representative DUT-49 materials, (**4**), (**11**), (**12**) and (**14**) with average crystal sizes of 150 nm, 850 nm, 1.16 μm and 4.27 μm, respectively, were selected and high-resolution *n*-butane isotherms at 298 K were recorded (Fig. 1d). All isotherms share the same shape with a distinct step at 30 kPa caused by DUT-49 contraction. While the experimental uptake of (**4**) after and before the structural transition at 35 kPa is in very good agreement with the value predicted by grand canonical Monte Carlo (GCMC) simulations for the crystal structures of DUT-49*op* and DUT-49*cp*, decreasing crystal size leads to a gradual drop of the *n*-butane amount adsorbed which corresponds well to the observations during nitrogen adsorption at 77 K. In situ PXRD in parallel to the adsorption of *n*-butane at 298 K (Supplementary Fig. 29) of these 4 representative samples clearly support the proposed mechanism. In all samples a transition of DUT-49*op* to DUT-49*cp* is observed in agreement to the previously reported data[23]. Only traces of

DUT-49*op* could be detected in the smaller crystallite samples after the NGA step, indicating that the transition does not occur in all crystals.

Interestingly, the magnitude of the NGA step as well as the PA given as $\Delta n_{NGA}$ and $\Delta p_{NGA}$, respectively, grow with increasing crystal size, clearly indicating that DUT-49 samples containing larger crystallites provide significantly higher pressure amplification under these conditions (Fig. 3). This effect is not directly proportional to the lower uptake of the samples, as DUT-49(**4**) has a twofold capacity compared with DUT-49(**14**), but almost ten times higher values of $\Delta p_{NGA}$ and $\Delta n_{NGA}$. This fact clearly indicates an increase of the activation energy for the NGA contraction with increasing crystallite size, a characteristic signature of cooperative phenomena such as ferromagnetism or ferroelectricity[28,29]. The concerted rotation of MOPs and simultaneous contraction of unit cells must proceed as a highly correlated process at a propagation rate of the order of a lattice vibration. In larger crystals switching of internal domains in opposite directions (micro-twinning) causes internal built-up of stress associated with energetic barriers. Alongside with downsizing, these activation energies decrease and vanish for monodomain phase transitions, a phenomenon reminiscent of ferromagnets that lose their remanent magnetization below a critical particle size to become superparamagnetic. However, the structural transformations are not totally suppressed per se in crystals below a critical size as even the smallest particles contract in the presence of *n*-butane (Fig. 1d).

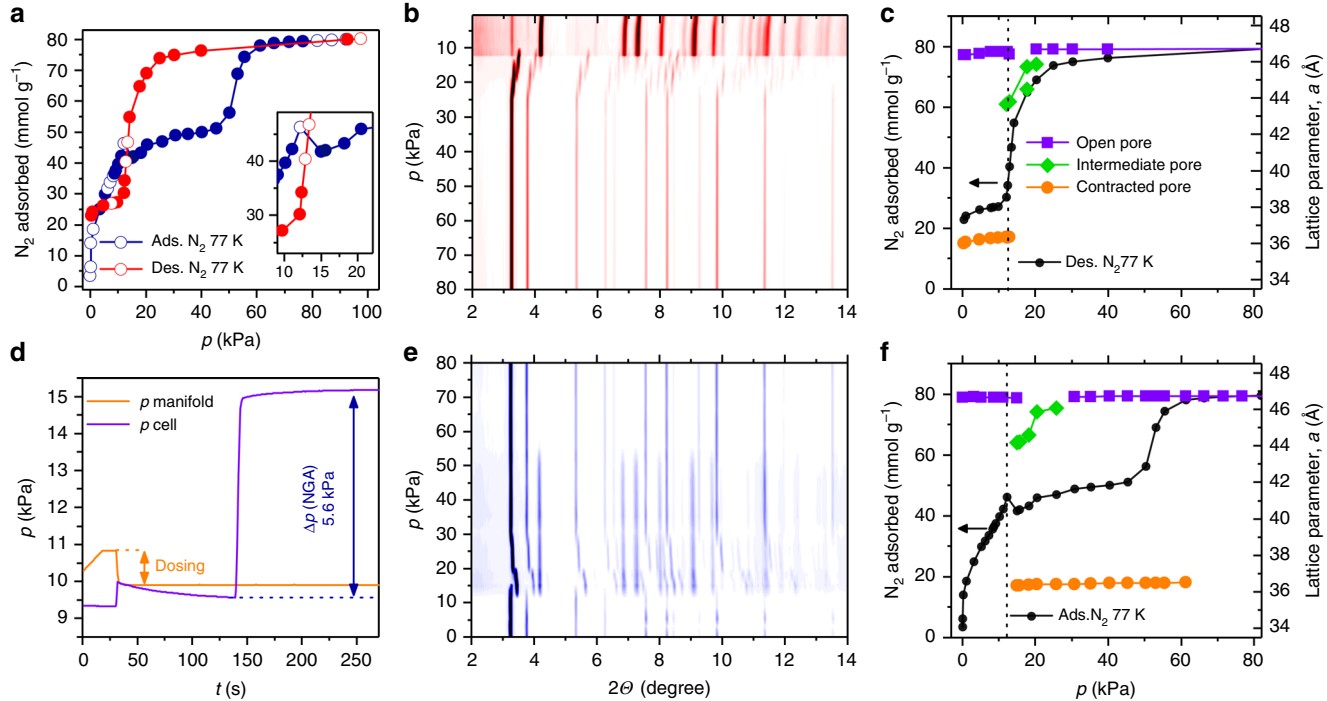

**Fig. 2** In situ PXRD in parallel to nitrogen adsorption. **a** Adsorption isotherm of nitrogen at 77 K of DUT-49(**4**) (adsorption points corresponding to collected PXRD patterns are represented as solid symbols) and **d** corresponding kinetic profile of the NGA transition. **b, e** Contour plot of in situ PXRD patterns obtained during desorption (red, **b**) and adsorption (blue, **e**). **c, f** Lattice parameter evolution during desorption (**c**) and adsorption (**f**). op: open pore phase (cubic, $Fm\overline{3}m$, $a = 46.368(20)–46.732(1)$ Å), cp: contracted pore phase (cubic, $Pa\overline{3}$, $a = 36.020(1)–36.504(16)$ Å), ip:intermediate pore phase (cubic, $Pa\overline{3}$, $a = 43.649(1)$ Å–$46.051(1)$ Å)), adsorption and desorption branch from (**a**) are represented as filled black symbols

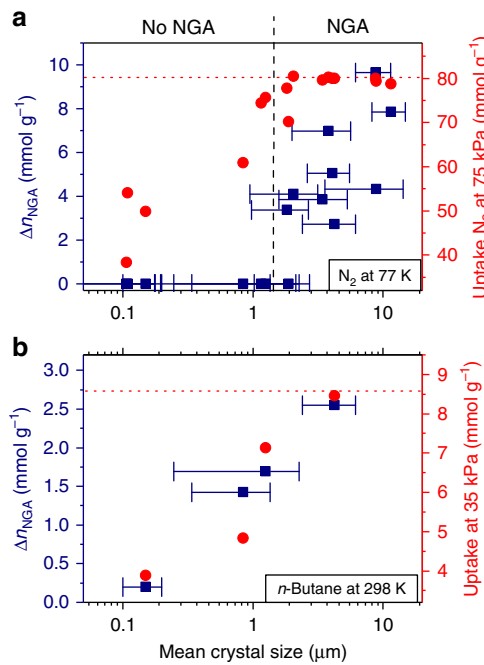

**Fig. 3** Relationship between $\Delta n_{NGA}$ and crystal size. Semi-logarithmic plot derived from **a** nitrogen adsorption at 77 K and **b** n-butane adsorption at 298 K. Values for $\Delta n_{NGA}$ are given as blue squares including error bars for crystal size standard deviation, total uptake for nitrogen and n-butane at 75 and 35 kPa, respectively, are given as red circles, and theoretical uptake is given as dotted red line

**Thermodynamics analysis.** According to our findings, the guest-induced contraction (and breathing) vanishes below a critical size because the GCMC-simulated adsorption enthalpy difference of DUT-49op and DUT-49cp ($\Delta\Delta H = \Delta H_{op} - \Delta H_{cp}$) for the guest molecule is insufficient in magnitude as it is observed for nitrogen here ($\Delta H_{op} = -10$ kJ mol$^{-1}$, $\Delta H_{cp} = -13$ kJ mol$^{-1}$, $\Delta\Delta H = -3$ kJ mol$^{-1}$, Supplementary Table 15), whereas the much higher |$\Delta\Delta H$| simulated for n-butane[23] ($\Delta H_{op} = -23.9$ kJ mol$^{-1}$, $\Delta H_{cp} = -47.9$ kJ mol$^{-1}$, $\Delta\Delta H = -24$ kJ mol$^{-1}$) enforces the contraction regardless of the particle size. The reduced |$\Delta\Delta H$| renders nitrogen as a significantly more sensitive probe to detect subtle energetic particle size-dependent phenomena, while the activation barrier variation observed using n-butane as a probe reflects the barrier for cooperative switching of an increased number of unit cells resulting in a monotonic barrier increase with increasing crystal size. The suppression of adsorption-induced transitions below a critical crystal size is a novel finding for pressure-amplifying solids. The size dependence of guest-induced flexibility in MOFs has only very recently been recognized and discussed for the cases of ZIF-8[30–33] and DUT-8[34]. Such tailoring of crystal sizes for flexible MOFs has been already recognized as a crucial basis for the design of porous shape memory materials by some of the pioneers in the field who also reported flexibility suppression in mesosized, interpenetrated MOF structures but an increasing hysteresis width for smaller crystals and concluded a suppression of phase transitions caused by an increased energy barrier for the structural transformation for the smaller crystals due to a decreased number of lattice defects[35]. In contrast, in DUT-49 the phase transformation barrier and $\Delta n_{NGA}$ decreases with decreasing particle size due to a lower transformation energy barrier for smaller crystals indicating the predominance of domain and grain boundary formation energetics in phase transitions associated with such gigantic volume changes.

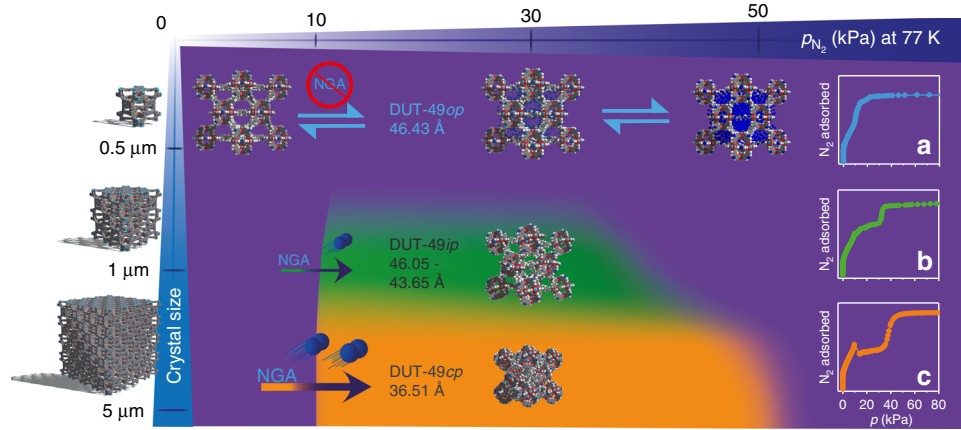

**Fig. 4** Phase transitions of DUT-49 depending on crystal size and pressure during adsorption of nitrogen at 77 K. Structural representations are given for the observed phases corresponding to unit cell parameters given in the caption of Fig. 2. Nitrogen adsorption isotherms at 77 K for **a** DUT-49(**12**), **b** DUT-49(**10**) and **c** DUT-49(**4**) representing the three prototypical scenarios

**Adsorption-induced switching mechanism.** A more in-depth analysis of the isotherms and in situ PXRD data for nitrogen reveals insightful details of the switching mechanisms for the two probe molecules. While a plateau is observed after the NGA step in the *n*-butane isotherms at 298 K (also observed for methane[23] and xenon[25]), the shape of the nitrogen isotherm at 77 K for larger crystallite samples such as DUT-49(**4**) (4.26 μm) is stepwise before the reopening of the structure at 45–50 kPa (Fig. 2a). This, and the fact that the uptake of nitrogen after NGA exceeds the theoretical uptake of DUT-49*cp* predicted by GCMC simulations by more than 10 mmol g$^{-1}$, indicates a more complex switching mechanism upon adsorption of nitrogen at 77 K (Fig. 4).

Thus, we analyzed DUT-49(**4**) (4.26 μm) by in situ PXRD (Fig. 2) during nitrogen adsorption at 77 K. Interestingly, DUT-49*op* does not entirely transform into DUT-49*cp* at 15.02 kPa but into a mixture of DUT-49*op*, DUT-49*cp* and an intermediate phase (DUT-49*ip*). At higher pressures the unit cell volume of the *ip* phase increases continuously and finally transforms into DUT-49*op* at 30.7 kPa. The *cp* phase, however, remains unchanged up to a pressure of 50 kPa, where it also transforms into DUT-49*op*. The observed structural transitions confirm a multi-step adsorption mechanism reflected by the isotherm shape with a higher uptake in the range of 10–50 kPa compared to the value predicted by GCMC simulations for pure DUT-49*cp*, while the initial and final adsorption regimes are very well reproduced by considering the contribution of DUT-49*op* (Fig. 1c). Increasing the crystal size clearly causes a shift towards higher fractions of *cp*, and *ip* vs. *op* phases in mixtures present at intermediate pressure (Fig. 2). We also observe an increase in total uptake with increasing crystal size upon adsorption of nitrogen at 77 K, similar to isotherms recorded for *n*-butane at 298 K (Fig. 1). In contrast, the smallest crystals did not show any hysteresis upon adsorption of nitrogen at 77 K due to the persistence of DUT-49*op* throughout the whole pressure range (Fig. 4). Only crystals above a critical size of 1 μm undergo a transition in either *cp* or *ip* phase. Because all samples consist of particles with certain size distribution, it is likely that smaller crystals in the sample retain the *op* structure while larger crystals transform into either *ip* or *cp*. To prove this hypothesis we performed the same in situ experiment on DUT-49(**10**), a sample with smaller hysteresis and an average crystal size of 1.2 μm (Supplementary Fig. 27). A transition takes place at 13.1 kPa, but no transformation into DUT-49*cp* could be observed. Instead, a phase mixture of several *ip* phases could be detected, which explains the smaller hysteresis and lower $\Delta n_{\mathrm{NGA}}$ value. The presence of *ip* phases upon adsorption in both cases is

unprecedented in DUT-49 and results from the weaker interaction of nitrogen with the framework as compared to other gases investigated. Because both the degree of the structural contraction and the magnitude of the distortion of the ligand are lower in an *op*–*ip* transition compared to an *op*–*cp* transition, referring to previous findings[24] the energy required to compensate this structural deformation is thus expected to be lower[24], and hence the subtle crystal size-dependent energetic changes are revealed throughout the transition. Upon nitrogen desorption, both DUT-49(**4**) and (**10**) show transformation into an *ip* phase in the pressure range of 21–17 kPa, followed by a mixture of DUT-49*cp* and DUT-49*op* at lower pressures. As far as we are aware of, this is the first report of a particle size-dependent observation of new intermediate structures in flexible MOFs.

## Discussion

The herein described observations reveal for the first time a systematic tuning of porous, pressure-amplifying materials by particle size engineering. An early thermodynamic explanation for flexibility suppression below a critical particle size has been given by recognizing a reduced density of the adsorbed phase in the outer crystal layer which diminishes the driving force ($|\Delta\Delta H|$) for the adsorption-induced deformation[32,33]. However, in the case of single-step gate opening, this theory cannot explain the stabilization of an open phase in the absence of guests[34]. An equally important but to a less extent understood contribution counteracting the adsorption-induced deformation is expected to result from an increasing dominance of the outer surface area ($A_{\mathrm{o}}$) and its corresponding interfacial energy for individual crystals in contact with a surrounding matrix generated either by other crystals, amorphous side products or a surface layer (Supplementary Fig. 41). Such matrix-induced stiffening is expected to equally suppress flexibility below a critical particle size. In contrast, the observed reduced PA is a characteristic of the reduced activation energy for the crystal transformation and unrestricted fluid–solid equilibration of smaller crystals. In large crystals of DUT-49*op* the massive volume change causes internal stress and the formation of multiple domains, internal grain boundaries associated with an interfacial domain area ($A_{\mathrm{id}}$)[34], a process associated with a high activation energy. In contrast, below a critical particle size, a mono-domain transformation will require no or only marginal activation energy for grain boundary formation and thus the disappearance of NGA is expected which is nearly the case here (Fig. 1d, e).

Thus, by tailoring the crystallite size of DUT-49 from 0.1 to over 10 μm, we have demonstrated NGA for nitrogen adsorption to depend critically on particle size, and below 1–2 μm the structural contraction is suppressed (Fig. 3). In contrast, NGA upon adsorption of *n*-butane at 298 K was observed for micrometer as well as nanometer scaled crystals, showing that NGA can be used in thin films and miniaturized devices. Larger crystals (>4 μm) induce higher PA due to the higher activation barrier associated with cooperative switching of a large number of unit cells, while for 150 nm-sized particles the barrier almost vanishes. As revealed by parallelized adsorption/synchrotron diffraction experimentation, particle sizing alters the complex structural transformation mechanism and reveals unprecedented intermediates along multiple structural transformations in DUT-49 upon adsorption of nitrogen at 77 K. Repeated PA initiated by nitrogen at 77 K showed no sign of sample degradation over multiple cycles (Supplementary Fig. 30, 31). These results demonstrate tuning of NGA by tailoring the crystal size of ordered porous networks. The experimental data underpin the complexity and uniqueness of the observed transitions among other porous materials and reveal the importance of particle size-dependent investigations for switchable porous materials. We hope our observations may serve as a stimulus for the future advancement of multi-scale computational methods heading towards a more quantitative analysis of domain and crystal size-dependent solid-phase energetics in nanoporous solids.

## Methods

**Materials**. All materials and gases used in the synthesis and analysis of DUT-49 samples were of high purity. $Cu(NO_3)_2 \cdot 3H_2O$ (Sigma Aldrich, 99.5%), *N*-methyl-2-pyrrolidone (NMP) (AppliChem, 99%), anhyd. deuterated dimethyl sulfoxide (DMSO; Sigma Aldrich, 99.9%), DCl in $D_2O$ 35 wt.% (Sigma Aldrich, 99%) and anhyd. ethanol (VWR Prolabo, 99%) were used for the synthesis of DUT-49 ($C_{40}H_{20}N_2O_8Cu_2$). The ligand $H_4BBCDC$ (9,9′-([1,1′-biphenyl]-4,4′-diyl)bis(9*H*-carbazole-3,6-dicarboxylic acid)) was synthesized following a procedure previously reported by our group. The synthetic protocol and analytic data are available in the literature[26].

**Synthesis of DUT-49 samples**. Similar to the previously published procedure for the synthesis of DUT-49[23], the synthesis of DUT-49 samples in this work is based on a solvothermal reaction of $H_4BBCDC$ with 2.1 equivalents of $Cu(NO_3)_2 \cdot 3H_2O$ in NMP at 80 °C. To obtain a set of 16 samples (referred to as DUT-49(**1**) to DUT-49(**16**)) with different crystal size distributions, the synthetic parameters such as ligand concentration, reaction time and nature and amount of modulator (acetic acid (AcOH) to increase crystal size and triethylamine (NEt₃) to decrease the crystal size) were varied (Supplementary Tables 1–3 for reaction conditions).

$H_4BBCDC$ (100 mg, 0.151 mmol) was reacted with $Cu(NO_3)_2 \cdot 3H_2O$ (76.8 mg, 0.317 mmol) in PYREX tubes 25 ml in volume. The ligand was dissolved in NMP at 60 °C and increasing amounts of acetic acid were added followed by the addition of $Cu(NO_3)_2 \cdot 3H_2O$ to the solution at 60 °C. The sealed tubes were placed in an oven at 80 °C for the specified reaction time.

The synthesis of sample DUT-49(**4**) (mean crystal size 4.26 μm) was scaled up using 1.250 g of $H_4BBCDC$ ligand, 0.957 g $Cu(NO_3)_2 \cdot 3H_2O$, 16.2 ml acetic acid and 213 ml NMP. The reaction was performed in a SCHOTT bottle (250 ml) instead of PYREX tubes for 48 h at 80 °C. The following sample treatment was identical for all other samples. All analysis data given for sample DUT-49(**4**) in this work are based on the scale-up synthesis.

Due to the fast reaction and precipitation upon addition of triethylamine (NEt₃), the reaction conditions were adjusted (Supplementary Table 3): to ensure good mixing of the reaction components, a solution of $H_4BBCDC$ (100 mg, 0.151 mmol) in NMP (15.2 ml) was placed into a flask and stirred at 80 °C. Cu(NO₃)₂·3H₂O (76.8 mg, 0.317 mmol) was added to the solution and dissolved followed by the addition of NEt₃. To investigate the impact of stirring on the reaction, sample **13** was synthesized without addition of NEt₃.

**Activation of DUT-49 samples**. After the solvothermal reaction the blue precipitates were separated from the reaction solution by centrifugation and washed 6 times with fresh NMP over a period of 2 days at room temperature. NMP was exchanged by anhyd. ethanol by washing with fresh ethanol 10 times over a period of 4 days. All samples were subjected to an activation procedure involving supercritical $CO_2$, as previously described for DUT-49[23]. After suspension in ethanol the samples were placed on filter frits in a Jumbo Critical Point Dryer 13200J AB (SPI Supplies) which was subsequently filled with liquid $CO_2$ (99.995%

purity) at 15 °C and 50 bar. To ensure a complete substitution of ethanol by $CO_2$, the liquid in the autoclave was exchanged with fresh $CO_2$ 18 times over a period of 5 days. The temperature and pressure were then raised beyond the supercritical point of $CO_2$ (35 °C and 100 bar) and kept until the temperature and pressure was constant at least for 1 h. Supercritical $CO_2$ was slowly released over 3 h and the dry powder was transferred and stored in an argon-filled glove box.

The influence of varying activation procedures on the adsorption behavior was investigated on sample DUT-49(**4**) (mean crystal size 4.26 μm). Samples suspended in ethanol were activated with supercritical $CO_2$ according to the previously described procedure, but time of soaking in liquid $CO_2$ was varied from 16 h to 5 days. In addition, activation at elevated temperatures from 80 °C to 200 °C for ca. 16 h was performed on the supercritically activated sample that was soaked in $CO_2$ for 5 days. For thermal activation, 100 mg of sample was placed in a Schlenk tube and connected to dynamic vacuum. After evacuation at room temperature for 1 h the tube was heated to the temperatures indicated in Supplementary Fig. 21.

**Characterization**. Analyses were carried out on the fully activated samples (supercritically activated over 5 days and additionally thermally activated in dynamic vacuum ($p < 0.1$ Pa) at 150 °C). Only PXRD measurements were performed on solvated, as made samples, proving the phase purity of the synthesized powders.

NMR analysis was performed on samples after activation, prior and after nitrogen adsorption experiment: 8 mg of each sample was suspended in 700 μl anhyd. deuterated DMSO, 50 μl of DCl in $D_2O$ (35 wt.%) were added and the NMR tube was sealed. The suspension was treated in an ultrasonic bath at 40 °C for 20 min and $^1H$ NMR spectra were recorded on a BRUKER DRX-500 (500.13 MHz).

For routine phase analysis, the synthesized and activated DUT-49 samples were analyzed by PXRD. Diffraction patterns were collected in transmission geometry with a STOE STADI P diffractometer operated at 40 kV and 30 mA, with monochromatic Cu-K$_{\alpha1}$ ($\lambda = 0.15405$ nm) radiation, a scan speed of 20 s per step and a step size of 0.1° 2Θ. Activated samples were prepared under inert atmosphere.

TGA was carried out in synthetic dry air using a NETZSCH STA 409 thermal analyzer at a heating rate of 5 K min$^{-1}$.

SEM images of DUT-49 samples (prepared in argon) were taken with secondary electrons in a HITACHI SU8020 microscope using 1.0 kV acceleration voltage and 10.8 mm working distance. Samples DUT-49(**14**) to (**16**) were sputtered with a 10 nm thick gold layer to avoid charge built-up. For each sample a series of images was recorded at varying magnifications and for each sample three different spots on the sample holder were investigated. The crystal size presented in this study refers to the edge length of the cubic DUT-49 crystals. The analysis of the SEM images was performed with ImageJ Software package[36]. Values for mean crystal size, as well as standard deviation in nm (SD) and relative standard deviation in percent were obtained using the ImageJ Analyze-Distribution function.

Volumetric adsorption experiments were carried out on a BELSORP-max instrument using gases of high purity (nitrogen: 99.999%, *n*-butane: 99.95%, He: 99.999%). For volumetric adsorption experiments the measuring routine of BELSORP-max was used. Targeted relative pressures in the range of 0.01–100 kPa were defined and limits of excess and allowance amount were set to 10 and 20 cm³ g$^{-1}$, respectively. For high-resolution isotherms of *n*-butane limits of excess and allowance amount were set to 2 and 4 cm³ g$^{-1}$, respectively. Equilibration conditions for each point were set to: 1% pressure change within 350 s. The dead volume was routinely determined using helium. Values for the adsorbed amount of gas in the framework are all given at standard temperature and pressure. Liquid nitrogen was used as coolant for measurements at 77 K and a Julabo thermostat was used for measurements at 298 K.

Elemental analyses of selected samples were carried out on a VARIO MICRO cube Elemental Analyzer by ELEMENTAR ANALYSATORSYSTEME GmbH in CHNS modus. Samples were analyzed after activation in dynamic vacuum at 150 °C (as used for the adsorption experiments). To avoid contamination with moisture all samples were sealed in the analysis capsules in a glove box under Ar atmosphere. For each sample three individual measurements were performed and the average was calculated as final result.

DRIFT spectroscopy was performed on selected samples on a BRUKER VERTEX 70 with a SPECAC Golden Gate DRIFT setup. Prior to the measurement, 2 mg of DUT-49 sample was mixed with 15 mg dry KBr in a mortar and pressed in the DRIFT-cell.

**Parallelized (in situ) diffraction–adsorption instrumentation**. The experimental setup and procedure used for in situ PXRD experiments in parallel to adsorption has been described in detail in previously published reports[23,37]. The experiments were performed at BESSY II light source, KMC-2 beamline of Helmholtz-Zentrum Berlin für Materialien und Energie using the recently established experimental setup[37]. The diffraction experiments were performed in transmission geometry using a sample holder with a thickness of 2 mm. The monochromatic radiation with energy of 8048 eV ($\lambda = 1.5406$ Å) was used for all experiments. The diffraction images were measured using a VÅNTEC-2000 area detector system (BRUKER) in the range of 2Θ = 2–50°. A synchrotron beam with dimensions of 0.5 × 0.5 mm was used for the experiments. Corundum powder with a crystallite size of 5 μm was used as an external standard. The image frames were integrated using Datasqueeze

2.2.9 software[38] and processed using Fityk 0.9.8 program[39]. The PXRD patterns measured at defined points of the nitrogen physisorption isotherm are shown in Supplementary Fig. 25–27. For sample DUT-49(**4**) complete adsorption–desorption nitrogen isotherms at 77 K were measured and PXRD patterns were recorded after equilibration at selected points of the isotherm in situ by means of an automated dosing procedure of BELSORP-max. In situ nitrogen adsorption on sample DUT-49(**10**) as well as in situ adsorption experiments using *n*-butane at 298 K on samples DUT-49(**4**), (**11**), (**12**) and (**14**) were conducted manually by setting defined pressures of interest, based on the ex situ measured isotherms. For all measurements equilibration times of the adsorption/desorption pressures of over 400 s were applied before a PXRD pattern was recorded.

**Refinement of PXRD patterns**. Profile matching of all PXRD was performed in the FullProf[40] software using the LeBail method[41]. A modified pseudo-Voigt function involving axial divergence asymmetry was used for the profile refinement[42]. Detailed analysis of the refinement is provided in supplementary notes and refinement parameters are provided in Supplementary Tables 6–8.

**GCMC simulations**. GCMC simulations were carried out at 77 K to predict the single component adsorption of nitrogen. The simulation boxes were made of a single unit cell ($1 \times 1 \times 1$) for all DUT-49 structural forms. Short-range dispersion forces described by Lennard-Jones (LJ) potentials were truncated at a cut-off radius of 12 Å. The fugacities for adsorbed species at a given thermodynamic condition were computed with the Peng–Robinson equation of state[43]. For each state point, $5 \times 10^7$ Monte Carlo steps were used for both equilibration and production runs. The adsorption enthalpy at low coverage ($\Delta H$) for each gas was calculated through configurational-bias Monte Carlo simulations using the revised Widom's test particle insertion method[44]. In order to gain insight into the configurational distribution of the adsorbed species in the open, intermediate and contracted forms of DUT-49, some additional data were calculated at different pressures including the radial distribution functions between the guests and the host, and the density probability of the adsorbates within the porosity (Supplementary Fig. 32–40). Details on the microscopic models for DUT-49 and interatomic potentials are provided in the supplementary discussion and Supplementary Tables 10–15.

**Microscopic models for DUT-49**. The crystal structures of DUT-49*op* and DUT-49*cp* were taken from our previous study[23] and they correlate well to the structural transitions observed in this work. Structures of DUT-49*ip-ads* and DUT-49*ip-des* were refined from the PXRD patterns displayed in Supplementary Fig. 27. The partial charges for each atom of the DUT-49 framework were derived at the density functional theory (DFT) level using a Mulliken partitioning method[45]. This calculation was performed by means of the CRYSTAL14 software package[46] with the use of a single-point energy calculation on the crystal model of DUT-49*op* previously DFT optimized[23]. The PBEsol generalized gradient approximation exchange–correlation functional (Perdew–Burke–Ernzerhof (PBE) revised for solids)[47], and all-electron basis sets for all atoms were applied[48] and the dispersion interactions were accounted for by using the Grimme "D2" dispersion correction scheme[49]. The corresponding atomic partial charges are summarized in Supplementary Fig. 32 and Supplementary Table 10. The same charges were considered for the intermediate and contracted forms of DUT-49 since the chemical environments for each atoms are the same in all these structures. Complementary to this, the partial atomic charge analysis on DUT-49 was further performed using REPEAT method proposed by Campañá et al.[50]. which was recently implemented into the CP2K code[51–54]. These calculations used the PBE functional[55] along with a combined Gaussian basis set and pseudopotential available in the CP2K package[51–54]. In the case of carbon, oxygen, nitrogen and hydrogen, a triple zeta (TZVP-MOLOPT) basis set was considered, while a double zeta (DZVP-MOLOPT) was applied for copper[56]. The pseudopotentials used for all atoms were those derived by Goedecker et al.[57]. The van der Waals interactions were taken into account via the use of semi-empirical dispersion corrections as implemented in the DFT-D3 method[58]. We also employed the charge equilibration method (QEq) proposed by Rappe and Goddard[59] to extract another set of charges. All corresponding atomic partial charges are summarized in Supplementary Fig. 32 and Supplementary Tables 11–14. The same charges were considered for the intermediate and contracted forms of DUT-49 since the chemical environments for each atom are similar in all these structures.

**Interatomic potentials**. The interactions between the DUT-49 frameworks and the guest species ($N_2$) were modeled using the sum of a 12–6 LJ contribution and a coulombic term. The Universal force field (UFF)[44] and DREIDING[60] were considered to describe the LJ potential parameters for the atoms of the inorganic and organic part of the DUT-49 framework, respectively (Supplementary Tables 11–14). In this work, $N_2$ was represented by a three-site charged model with two LJ sites located at the N atoms while a third site present at its center of mass only involves electrostatic interactions as previously described in the TraPPE force field[61] The LJ parameters for the guest/host interactions were obtained using the Lorentz-Berthelot combination rules.

**Data availability**. The data that support the findings of this study are available from the corresponding author.

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

## Acknowledgements

This project has received funding from the European Research Council (ERC) under the European Union's Horizon 2020 research and innovation programme (grant agreement no 742743). The authors thank the BMBF (no. 05K16OD3) and ANR/DFG Program FUN for financial support and Helmholtz-Zentrum Berlin für Materialien und Energie for allocated beam time and travel funding. The authors are grateful to Sebastian Ehrling and Ulrike Koch for SEM images. The authors want to especially thank J. Evans and F.-X. Coudert for helpful discussions and for providing the charges of DUT-49. G.M. thanks Institut Universitaire de France for its support.

## Author contributions

S. K. synthesized, activated and performed characterization of DUT-49 samples. S. K., V. B., D. W. and D. M. T. contributed to in situ PXRD measurements. R. S. P. and G. M. contributed to simulation of adsorption isotherms. S. K., V. B., I. S. and S. Kaskel contributed to analysis, interpretation and discussion of adsorption and diffraction data. S. K., V. B., I. S., R. S. P., G. M. and S. Kaskel contributed to writing the manuscript.

## Additional information

**Competing interests:** The authors declare no competing interests.

