## [Peer Review File · Nature Communications]

Reviewers' comments:

Reviewer #1 (Remarks to the Author):

This manuscript reports the control of the MOF crystal size can be used to modulate the negative gas adsorption behaviour. The study looks at the specific case of DUT-49.

The manuscript is well-written and well-referenced. The results reported are intriguing and supported by a number of complementary analyses. The findings are novel and important for the field. I would therefore recommend publication after minor revisions as noted below:

- The motivation is not made quite clear. What prompted the authors to specifically investigate the effect of crystal size on pressure amplification?
- It is not clear what Fig 1e shows;
- Page 6, line 151: ZIF-8 AND DUT-8
- Is there a way to generalise the findings to other MOFs for a given adsorbate? To different adsorbates for a given MOF?

Reviewer #2 (Remarks to the Author):

This manuscript shows interesting negative gas adsorption phenomena that depend critically on crystallite size. The work is very thoroughly carried out and well described. Although the MOF itself and the NGA phenomena have been reported before, the dependence on crystallite size is new. In this light, I feel the manuscript must do a better job of contextualizing their work. There is mention of other works that show downsizing but really no comparative discussion particularly the Kitagawa Science paper in 2013. In its present form, for a general journal as Nature Comm, the reader is presumed to know the context and this may not be accurate. I would offer the authors a chance to expand the discussion on the downsizing and make it more comparative as this is the only novel aspect in this paper. As it is, the novelty is not adequately conveyed to merit acceptance.

Reviewer #3 (Remarks to the Author):

Despite a certain degree of hype surrounding Metal-Organic Frameworks (MOFs), there have been some interesting new physics arising from studies of adsorption in this class of materials. One of these is framework flexibility, also termed gate-opening effects. In some extreme cases, as shown by some of the authors of the present paper, this can lead to release of adsorbed gases upon increase in pressure – a phenomenon described as negative gas adsorption (NGA), a terminology which I will accept for lack of a better alternative. This is a rather counterintuitive phenomenon, and as such deserves more in-depth research to clarify its governing mechanism, as well as identify any controlling factors. The paper under review presents such a study, where a combination of experimental and theoretical methods is used to shed new light into the mechanism of NGA in the DUT-49 material. The study is at least partially successful, and I would thus recommend its publication in Nature Communications. However, I feel some issues should be explored or clarified further before publication:

1- From a formal point of view, the figures contain a large amount of information that is sometimes difficult to process. This can be improved, in my opinion, by providing additional information in the figures themselves and in the figure captions. For example, in Figures 1a and 1b, the four images/graphs should be labelled with the respective sample code (as per Tables S1-S3); in Figure 1c, the meaning of the closed and open symbols is not provided. This is meant as an example, and I recommend that the authors revise their figures and captions to ensure all the necessary information is at hand to enable full understanding of the meaning and objective of each figure.

2- The authors argue that all samples correspond to the pure DUT-49 phase. However, some samples (typically those with smaller crystals) show a significant decrease in the total adsorption capacity – see e.g. Figures 1d and 1f. As far as I could understand, the authors do not provide a convincing explanation to this observation. Moreover, is the decrease in NGA in any way associated to this decrease in capacity? If so, why? On a related note, agreement between simulation and experiment is quite good for the samples with large crystals but deteriorates as the crystal size decreases (see Figures S1-S16). Could this be another indication of samples becoming progressively less pure?

3- It is clear that there is a correlation between the crystal size and the magnitude of NGA, but this is never presented as such, just inferred qualitatively. It would be interesting to plot the amount of NGA against the average crystallite size directly, to see if a clear trend emerges. It appears to me that the change is non-monotonic, and if so that deserves further explanation.

4- Examining Tables S1-S3, it is hard to discern clear cause-effect relations between the synthesis conditions and the size of the crystals. For example, sample 3 has much higher concentration of AcOH than sample 2, but almost the same crystal size. Sample 10 has much higher concentration of NMP and twice the reaction time as sample 9 (both of which should lead to larger crystal formation), but significantly lower crystal size. I realise that controlling the crystal size of this type of material is highly challenging, but this raises questions about the reproducibility of the results. It would be important for the authors to provide an estimate of the uncertainty of their results, particularly with regards to the properties of each sample of material. Notice that I am not talking about a measure of the spread of the crystal size distribution for a particular sample, which is provided, but rather a measure of the statistical uncertainty in, say, the average crystal size from sample to sample. In this context, a comparison between samples 4 and 6 may be insightful – they seem to correspond to identical synthesis conditions, but show a difference of more than 10% in the average crystal size. The authors' conclusions should always be interpreted in terms of this statistical variability.

5- The authors offer an explanation for the observed NGA phenomenon based on changes to the activation barrier for the structural transition with crystal size. They also make use of energetic considerations to explain the mechanism of the structural transition. However, considerations of entropy changes during the process appear to be absent from this hypothesised mechanism. This being a solid phase, perhaps they are small enough to be negligible, but this should be commented upon. Furthermore, can the authors make some effort to roughly quantify the energetic components that play a role in the transition? This would make their argument much stronger, I believe. For example, at the top of page 3, the authors argue that the energy of compression of the structure is compensated by the difference in adsorption enthalpy in the two forms of the solid. However, in NGA there is also release of gas, that carries with it enthalpy and entropy changes. Do these also play a role in the free energy balance of the system? In another case, near the middle of page 7, the authors argue that the energy difference of the op-ip transition is lower than that of the op-cp transition. Has this been quantified? If not, what is the authors' basis for this statement?

6- The phenomenon of "cooperative switching" of unit cells of the structure (pages 5 and 6) deserves further explanation. It was not obvious to me how this is meant to operate, and it appears to be crucial to explain the effect of crystal size.

7- If I understood correctly, the authors argue that within a given sample, characterised by a distribution of crystal sizes, some of the crystals may be undergoing a transition, while others do not. This makes sense, and qualitatively explains the isotherm behaviour. However, given that they authors have measured the crystal size distribution, and have calculated GCMC isotherms for each of the structure types, would it not be possible for them to reconstruct the experimental adsorption isotherms as a composite of the simulated isotherms with the percentage of pores undergoing each transition? For example, if a sample is composed of 50% of crystals below the

op-cp threshold at a certain pressure, then its isotherm will be the 50/50 weighted average of the op and cp GCMC isotherms. Of course, in reality the picture is complicated by the presence of several intermediate phases, at least in some samples. Nevertheless, it might be useful to attempt this kind of calculation, as it would provide more quantitative evidence for the authors' arguments.

8- On a related topic, why is the GCMC isotherm for the op form not shown in Figure 1d?

9- As mentioned above, agreement between simulation and experiment is quite good, at least for samples with large crystals. However, the authors made use of Mulliken charges for the framework atoms. Mulliken charges are known to be highly unreliable, changing significantly with basis set, DFT functional, etc, and tend to provide a poor representation of the underlying electrostatic potential, which is an essential quantity for classical simulations of the type performed here. It would thus be important to assess how dependent the simulation results are to the choice of point charges.

Reviewer #1 (Remarks to the Author):

This manuscript reports the control of the MOF crystal size can be used to modulate the negative gas adsorption behaviour. The study looks at the specific case of DUT-49.

The manuscript is well-written and well-referenced. The results reported are intriguing and supported by a number of complementary analyses. The findings are novel and important for the field. I would therefore recommend publication after minor revisions as noted below:

- The motivation is not made quite clear. What prompted the authors to specifically investigate the effect of crystal size on pressure amplification?

The new NGA phenomenon, so far only reported for DUT-49, poses several open questions for future generalization. In order to make use of such pressure amplifying materials in near future, it is necessary to analyse the importance of molecular structure vs. mesoscopic structural effect. In order to compare the magnitude of NGA for different materials that might show similar behaviour in the future, this work clearly demonstrates that the crystal size effect cannot be neglected. A comment was added in the introduction (page 3, line 67) for clarification: "For a rational development of next generations of NGA materials enabling high pressure amplification, an understanding of critical size phenomena affecting metastability and NGA is essential."

- It is not clear what Fig 1e shows;

We included a sentence in the caption of figure 1:

pressure evolution in the cell upon NGA during *n*-butane adsorption at 298 K

In addition, we added the sample ID in Fig 1 a and included the symbol description: "Closed symbols correspond to adsorption, empty symbols to desorption"

- Page 6, line 158: ZIF-8 AND DUT-8

The word "and" was included at the appropriate position

- Is there a way to generalise the findings to other MOFs for a given adsorbate? To different adsorbates for a given MOF?

We inserted a new Fig. 3 into the manuscript showing the relation of crystal size and NGA for both N₂ at 77 K and *n*-butane at 298 K. In both cases an increasing $\Delta n(\text{NGA})$ with increasing crystal size is observed. This shows a generalization for different adsorbates for a given MOF structure (in this case DUT-49). We are currently on the way to identify new NGA MOFs with promising results, but so far DUT-49 represents the sole NGA material published to date and it would be speculative to generalize the findings for other MOFs. However, most probably other breathing mesoporous MOFs will follow a similar line, and examples of crystal size dependence in microporous flexible MOFs (such as ZIF-8) also show a decrease of hysteretic behaviour with decreasing crystal sizes similar to DUT-49.

Figure 3. Relationship between Δn_{NGA} and crystal size for a) N_2 adsorption at 77 K and b) n -butane adsorption at 298 K as semi-logarithmic plot. Values for Δn_{NGA} are given as blue squares including error bars for crystal size standard deviation, total uptake for N_2 and n -butane at 75 kPa and 35 kPa, respectively are given as red circles, theoretical uptake is given as dotted red line.

Reviewer #2 (Remarks to the Author):

This manuscript shows interesting negative gas adsorption phenomena that depend critically on crystallite size. The work is very thoroughly carried out and well described. Although the MOF itself and the NGA phenomena have been reported before, the dependence on crystallite size is new. In this light, I feel the manuscript must do a better job of contextualizing their work. There is mention of other works that show downsizing but really no comparative discussion particularly the Kitagawa Science paper in 2013. In its present form, for a general journal as Nature Comm, the reader is presumed to know the context and this may not be accurate. I would offer the authors a chance to expand the discussion on the downsizing and make it more comparative as this is the only novel aspect in this paper. As it is, the novelty is not adequately conveyed to merit acceptance.

Of course the pioneering work of S. Kitagawa addressing memory effects in size-controlled flexible MOFs should be emphasized. His work was already discussed briefly in the original manuscript. In order to further emphasize this important early finding we have expanded this section in the revised manuscript (page 6-7 from line 160 to line 172) discussing important differences observed for NGA materials.

Reviewer #3 (Remarks to the Author):

Despite a certain degree of hype surrounding Metal-Organic Frameworks (MOFs), there have been some interesting new physics arising from studies of adsorption in this class of materials. One of these is framework flexibility, also termed gate-opening effects. In some extreme cases, as shown by some of the authors of the present paper, this can lead to release of adsorbed gases upon increase in pressure – a phenomenon described as negative gas adsorption (NGA), a terminology which I will accept for lack of a better alternative. This is a rather counterintuitive phenomenon, and as such deserves more in-depth research to clarify its governing mechanism, as well as identify any controlling factors. The paper under review presents such a study, where a combination of experimental and theoretical methods is used to shed new light into the mechanism of NGA in the DUT-49 material. The study is at least partially successful, and I would thus recommend its publication in Nature Communications. However, I feel some issues should be explored or clarified further before publication:

1- From a formal point of view, the figures contain a large amount of information that is sometimes difficult to process. This can be improved, in my opinion, by providing additional information in the figures themselves and in the figure captions. For example, in Figures 1a and 1b, the four images/graphs should be labelled with the respective sample code (as per Tables S1-S3); in Figure 1c, the meaning of the closed and open symbols is not provided. This is meant as an example, and I recommend that the authors revise their figures and captions to ensure all the necessary information is at hand to enable full understanding of the meaning and objective of each figure.

We included a sentence in the caption of figure 1:

e) pressure evolution upon NGA during *n*-butane adsorption at 298 K

In addition, we added the sample ID in Fig 1 a and included the symbol description: "Closed symbols correspond to adsorption, empty symbols to desorption"

2- The authors argue that all samples correspond to the pure DUT-49 phase. However, some samples (typically those with smaller crystals) show a significant decrease in the total adsorption capacity – see e.g. Figures 1d and 1f. As far as I could understand, the authors do not provide a convincing explanation to this observation. Moreover, is the decrease in NGA in any way associated to this decrease in capacity? If so, why? On a related note, agreement between simulation and experiment is quite good for the samples with large crystals but deteriorates as the crystal size decreases (see Figures S1-S16). Could this be another indication of samples becoming progressively less pure?

The relationship between NGA and the total uptake is discussed below (section 3).

To compare the adsorption profile and show that the pore systems are identical in all samples all N₂ isotherms collected at 77 K were normalized at a pressure of 75 kPa. An additional Figure S 22 was included in the ESI:

Figure S22. Normalized nitrogen adsorption isotherms of DUT-49(1) – DUT-49(16) (based on the uptake at 75 kPa for each isotherm) in regular (a) and semilogarithmic plot (b); c) and d) magnifications of low pressure region of a) and b), respectively.

All isotherms share the same slope at pressures up to 10 kPa showing that the pore filling mechanism and porosity in the micro and small mesopore regimes is identical in all investigated samples.

A discussion about the relationship between internal and external surface was added to the ESI (see section 8, page 46). The latter illustrates this relationship and provides an explanation for the observed decrease in uptake in the crystal size range observed for this study:

8. Calculation of internal and external surface area

The ratio of external to internal surface area in DUT-49 was estimated as shown in Figure S41. The difference between internal and external surface grows significantly with crystal size (edge length). As a result, external surface distortion, pore blocking, and amorphous surface layers may play a larger role in smaller crystals and cause a reduced total uptake. In this work, samples with reduced crystal size below 2 μm show an increased clustering and intergrowth of the crystals while samples with larger crystals seem to form individual separated particles (also see SEM images Figure S1-S16). Such factors may impact the total uptake and pore volume of smaller crystals stronger than in larger crystals. Another aspect is the reduced density of the adsorbed phase in the outer layer causing an apparent reduction of

surface area by crystal downsizing as shown for microporous MOFs like ZIF-8. This effect and its particle size dependence in mesoporous MOFs needs further investigations in future.

Figure S 41. Relationship of internal and external surface area and number of unit cells to the crystal edge length in a cubic DUT-49 crystal. External surface area was calculated assuming a flat surface of the cubic crystal. The internal surface per unit cell is 171 nm² and was calculated based on the BET area of 5476 m² g⁻¹. Critical edge length for NGA is marked as dashed line.

However, the hint for such a decreased adsorbate density in the outer layer may be given in the adsorption isotherms for samples DUT-49(14)-(16). These isotherms do not reach a plateau and show a tendency towards IUPAC type II behaviour with a positive slope indicating that mesopores on the outer layer are filled at higher relative pressure than those located deeper inside the volume of DUT-49. A similar sloping is also detected for the smaller particles in *n*-butane isotherms.

3- It is clear that there is a correlation between the crystal size and the magnitude of NGA, but this is never presented as such, just inferred qualitatively. It would be interesting to plot the amount of NGA against the average crystallite size directly, to see if a clear trend emerges. It appears to me that the change is non-monotonic, and if so that deserves further explanation.

To better discuss the relation between crystal size and NGA we added a new Figure 3 in the manuscript:

Figure 3. Relationship between Δn_{NGA} and crystal size for a) N_2 adsorption at 77 K and b) n -butane adsorption at 298 K as semi-logarithmic plot. Values for Δn_{NGA} are given as blue squares including error bars for crystal size standard deviation, total uptake for N_2 and n -butane at 75 kPa and 35 kPa, respectively are given as red circles, theoretical uptake is given as dotted red line.

Both plots (for N_2 and n -butane) clearly show the increasing NGA values with increasing crystal size. They also show the same trend of decreasing total uptake with decreasing crystal size, however, in the region of NGA for N_2 -induced NGA increases with comparable static total uptake indicating that there is no direct connection between Δn_{NGA} and the total uptake of the material.

4- Examining Tables S1-S3, it is hard to discern clear cause-effect relations between the synthesis conditions and the size of the crystals. For example, sample 3 has much higher concentration of AcOH than sample 2, but almost the same crystal size. Sample 10 has much higher concentration of NMP and twice the reaction time as sample 9 (both of which should lead to larger crystal formation), but significantly lower crystal size. I realise that controlling the crystal size of this type of material is highly challenging, but this raises questions about the reproducibility of the results. It would be important for the authors to provide an estimate of the uncertainty of their results, particularly with regards to the properties of each sample of material. Notice that I am not talking about a measure of the spread of the crystal size distribution for a particular sample, which is provided, but rather a measure of the statistical uncertainty in, say, the average crystal size from sample to sample. In this context, a comparison between samples 4 and 6 may be insightful – they seem to correspond to identical synthesis conditions, but show a difference of more than 10% in the

average crystal size. The authors' conclusions should always be interpreted in terms of this statistical variability.

We rechecked the synthetic conditions and corrected that sample 2 and 4 were synthesized using a slightly higher solvent amount of 172 ml g^{-1} in contrast to sample 1,3,5,6. A statement was also added to the ESI to explain general trends for particle size engineering:

From our experiments we derive the following relationships:

Crystal size can be increased by using a higher dissolution of the ligand, using higher concentrations of acetic acid, extending the reaction time and stirring the reaction mixture.

Smaller crystals can be obtained by concentrating the reaction mixture, using no acetic acid, using a base such as Et_3N and using shorter reaction times.

The standard deviation which describes the dispersity of the size distribution is the lowest for samples synthesized using high concentrations of modulator (acetic acid or Et_3N) or for stirred solutions without the addition of modulator. Samples which were synthesized with highly diluted reaction mixtures (sample DUT-49(9), (10)) show a broad crystal size distribution.

To obtain a very narrow distribution of crystal sizes the nucleation should be controlled with a low concentration of nuclei and a continuous growth of the seeds with a low or negligible additional nucleation after the initial seed formation. In the samples with high SD (especially sample DUT-49(9), (10) with high dilution of the ligand) we assume that nucleation takes place continuously throughout the reaction time, consequently producing smaller crystals by renewed nucleation while previously formed seeds continue to grow.

An additional specification of SD and RSD was added in the ESI in section 4.4.

5- The authors offer an explanation for the observed NGA phenomenon based on changes to the activation barrier for the structural transition with crystal size. They also make use of energetic considerations to explain the mechanism of the structural transition. However, considerations of entropy changes during the process appear to be absent from this hypothesised mechanism. This being a solid phase, perhaps they are small enough to be negligible, but this should be commented upon. Furthermore, can the authors make some effort to roughly quantify the energetic components that play a role in the transition? This would make their argument much stronger, I believe. For example, at the top of page 3, the authors argue that the energy of compression of the structure is compensated by the difference in adsorption enthalpy in the two forms of the solid. However, in NGA there is also release of gas, that carries with it enthalpy and entropy changes. Do these also play a role in the free energy balance of the system?

In reference 24: Evans, J. D., Bocquet, L. & Coudert, F.-X. Origins of Negative Gas Adsorption. *Chem* 1, 873-886, doi:10.1016/j.chempr.2016.11.004 (2016) Evans et al. show that in the structural transition of the MOF/solid entropy plays a negligible role. In their paper free energy profiles are provided for the whole system including entropy and enthalpy calculations. The energetics for the transition were also described in detail in their work and also in the original NGA paper in reference 23: Krause, S. et al. A pressure-amplifying framework material with negative gas adsorption transitions. *Nature* 532, 348-352, doi:10.1038/nature17430 (2016).

To clarify this comment, the sentences on page 2 line 56 and page 3 line 59-60 were revised.

“The driving force for the contraction is energetic with a negligible entropic contribution.²⁴”

“....and hence induces the contraction by compensating the energy required for the empty host compression as demonstrated in our earlier work.²³”

The simulation of crystal-size dependence of the structural transition is an interesting question, but it would require meso- and/or macroscopic modeling to fully understand this point which is beyond scope of this paper.

In another case, near the middle of page 7, the authors argue that the energy difference of the op-ip transition is lower than that of the op-cp transition. Has this been quantified? If not, what is the authors' basis for this statement?

In reference 24: Evans, J. D., Bocquet, L. & Coudert, F.-X. Origins of Negative Gas Adsorption. Chem 1, 873-886, doi:10.1016/j.chempr.2016.11.004 (2016) Evans et al. show that the energy required for the structural transition of DUT-49 is mostly governed by the amplitude of deformation of the ligand. They show that a lower degree deformation requires less energy for the transition. Based on this statement, the following sentence on page 7 lines 199-201 was amended: “Because both the degree of the structural contraction and the magnitude of the distortion of the ligand are lower in an *op-ip* transition compared to an *op-cp* transition, referring to previous findings²⁴ the energy required to compensate this structural deformation is thus expected to be lower²⁴, hence the subtle crystal size dependent energetic changes are revealed throughout the transition.”

6- The phenomenon of “cooperative switching” of unit cells of the structure (pages 5 and 6) deserves further explanation. It was not obvious to me how this is meant to operate, and it appears to be crucial to explain the effect of crystal size.

We have added an explanation to clarify this point on page 5 lines 140-145:

The concerted rotation of MOPs and simultaneous contraction of unit cells must proceed as a highly correlated process at a propagation rate of the order of a lattice vibration. In larger crystals switching of internal domains in opposite directions (micro-twinning) causes internal built-up of stress associated with energetic barriers. Alongside with downsizing, these activation energies decrease and vanish for mono-domain phase transitions, a phenomenon, reminiscent of ferromagnets that lose their remanent magnetization below a critical particle size to become superparamagnetic.

7- If I understood correctly, the authors argue that within a given sample, characterised by a distribution of crystal sizes, some of the crystals may be undergoing a transition, while others do not. This makes sense, and qualitatively explains the isotherm behaviour. However, given that they authors have measured the crystal size distribution, and have calculated GCMC isotherms for each of the structure types, would it not be possible for them to reconstruct the experimental adsorption isotherms as a composite of the simulated isotherms with the percentage of pores undergoing each transition? For example, if a sample is composed of 50% of crystals below the op-cp threshold at a certain pressure, then its isotherm will be the 50/50 weighted average of the op and cp GCMC isotherms. Of course, in reality the picture is complicated by the presence of several intermediate phases, at least in some samples. Nevertheless, it might be useful to attempt this kind of calculation, as it would provide more quantitative evidence for the authors' arguments.

Thank you for this valuable suggestion! We had performed this analysis prior to the initial submission and compared the uptakes with simulated isotherms of different DUT-49 phases. However, we found this type of analysis was too vague to give an accurate analysis of the phase composition. In sample 4 which was investigated by in situ PXRD we estimate ca. 25% ($\pm 10\%$) ip phase and 75 % ($\pm 10\%$) cp phase after the transition. However, the gradual increase of unit cell size of the ip phase makes a quantitative analysis very difficult. For a reliable analysis we would need DFT kernels as applied in the calculation of pore-size distributions by DFT methods and the isotherms for different phases of DUT-49 would be the input for the analysis.

A kernel attempt to this problem will be very viable for this system and in general for phase mixtures in flexible MOFs and will be the topic of an independent study.

8- On a related topic, why is the GCMC isotherm for the op form not shown in Figure 1d?

Thank you for this valuable comment! We have included the op phase GCMC isotherm for DUT-49 butane adsorption at 298 K.

9- As mentioned above, agreement between simulation and experiment is quite good, at least for samples with large crystals. However, the authors made use of Mulliken charges for the framework atoms. Mulliken charges are known to be highly unreliable, changing significantly with basis set, DFT functional, etc, and tend to provide a poor representation of the underlying electrostatic potential, which is an essential quantity for classical simulations of the type performed here. It would thus be important to assess how dependent the simulation results are to the choice of point charges.

We all know that there are many approaches to extract the charges and we can argue that all methods (Mulliken, qEQ, ESP-ChelpG, Repeat, Bader,...) have pros/cons that make the choice of charges arbitrary. This is also true for the choice of the FF parameters to describe the vdw interactions between the host and the guest. We could have the same discussion on the relevance of a generic FF (DREIDING, UFF, OPLS,...) that is commonly employed in the MOF community to describe the vdw interactions. In this specific case, since the unit cell of DUT-49 is huge, we initially selected the Mulliken scheme to extract the charges. To address the point raised by the reviewer, we considered two additional sets of charges calculated using the Qeq and REPEAT approaches. This is described in section 7 of the SI. The GCMC simulations were further performed for the cp, lp and ip forms using the three sets of charges. The comparison of the simulated N₂-adsorption isotherms reported in Figure S34 clearly emphasizes that the choice of charges only slightly affects the results. More interestingly, once we switched off the coulombic interactions, it was clearly evidenced that the vdw interactions govern the adsorption of N₂ in DUT-49 and indeed makes the choice of the charges much less relevant than in other systems where the coulombic contribution is important.

REVIEWERS' COMMENTS:

Reviewer #3 (Remarks to the Author):

The authors have made a significant effort to address my comments and suggestions, and I believe this has strengthened the manuscript. In the few cases where the issues were not directly addressed, a clear and adequate justification was provided. I noticed only a few typos (e.g., "build-up of stress" instead of "built-up of stress"), but apart from checking and correcting these, I would recommend publication as is.